# Perinatal depression and implementation of the "Thinking Healthy program" support intervention in an impoverished setting of Lima, Peru: Assessment before and during the COVID-19 pandemic

## Research Article

COVID-19; depression; pregnant women; Thinking Healthy program; Peru

**Corresponding author:**
Margot Aguilar;
Email: maguilar_ses@pih.org

Margot Aguilar[1] , Carmen Contreras[1,2], Giuseppe Raviola[3,4,5],
Alejandra Sepúlveda[1], Maricielo Espinoza[1], Leydi Moran[1], Lourdes Ramos[1,9],
Jesús Peinado[1,8], Leonid Lecca[1,3,4], Gloria A. Pedersen[7] , Brandon A. Kohrt[7] and
Jerome T. Galea[3,6]

[1]Socios En Salud, Lima, Peru; [2]Harvard Global Health Institute, Harvard University, Cambridge, MA, USA; [3]Department of Global Health and Social Medicine, Harvard Medical School, Boston, MA, USA; [4]Partners in Health, Boston, MA, USA; [5]Department of Psychiatry, Massachusetts General Hospital, Boston, MA, USA; [6]School of Social Work, University of South Florida, FL, USA; [7]Department of Psychiatry and Behavioral Sciences, George Washington University, Washington DC, USA; [8]School of Medicine, Faculty of Health Sciences, Peruvian University of Applied Sciences – UPC, Lima, Peru and [9]Escuela Profesional de Tecnología Médica, Universidad Privada San Juan Bautista, Lima, Peru

## Abstract

Socios En Salud (SES) implemented the Thinking Healthy program (THP) to support women with perinatal depression before and during the COVID-19 pandemic in Lima Norte. We carried out an analysis of the in-person (5 modules) and remote (1 module) THP intervention. Depression was detected using PHQ-9, and THP sessions were delivered in women with a score (PHQ-9 $\geq$ 5). Depression was reassessed and pre- and post-scores were compared. In the pre-pandemic cohort, perinatal depression was 25.4% (47/185), 47 women received THP and 27 were reassessed (57.4%), and the PHQ-9 score median decreased from 8 to 2, $p < 0.001$. In the pandemic cohort, perinatal depression was 47.5% (117/247), 117 women received THP and 89 were reassessed (76.1%), and the PHQ-9 score median decreased from 7 to 2, $p < 0.001$. THP's modalities helped to reduce perinatal depression. Pregnant women who received a module remotely also reduced depression.

## Statement on the Thinking Healthy article

Perinatal depression is a mental health problem that 24% to 40% of women may experience during pregnancy. It significantly affects not only maternal mental and physical health but also the newborn in physical, mental, and cognitive aspects, with serious repercussions in its adult life. Therefore, the need to cover mental health care for this population is essential.

To strengthen the Peruvian health system, Socios En Salud implemented the WHO Thinking Healthy program, with the aim of reducing symptoms of depression in pregnant women in the community with the help of a Community Health Agent (ACS) trained in the north of Lima Peru. Upon implementation, the intervention was completed in 116/432 pregnant women in the third trimester, before and after isolation during COVID-19.

Before the isolation due to COVID-19, the ACS carried out 16 sessions in the homes of pregnant women. And during the isolation due to COVID-19, the first module (4 sessions) was implemented remotely. In both interventions, the ACS received supervision and follow-up from mental health professionals.

The program managed to benefit pregnant women and ACS. Pregnant women received follow-up sessions by ACS and mental health personnel, reducing levels of depression, while the ACS benefited from training in the program and strengthened their skills in managing depression during maternity.

The results of the investigation show that the remote and face-to-face intervention managed to reduce the symptoms of depression in pregnant women. Likewise, it generates the opportunity to discuss possible modifications to the session program and to continue investigating the effectiveness and quality of low-intensity psychosocial interventions delivered in low-resource countries. Another long-awaited result was the scaling of the program to the public sector, including THP to the Maternal Mental Health Program to health professionals which is being led by the Ministry of Health.

## Introduction

Perinatal depression is one of the most prevalent mental health conditions in pregnant women worldwide (Miguez and Vazquez, 2021). Perinatal depression occurs in approximately 1 in 6 pregnant women (Austin et al., 2008; Austin et al., 2008), and prevalence estimates of perinatal depression range from 7 to 15% in high-income countries, although in low- and middle-income settings, as Latin America, perinatal depression is very high with an estimated pooled prevalence of 25% (Gelaye et al., 2016). For example, a study in a population of pregnant Latinas in the United States reportedly found that 32.4% had depression (Lara et al., 2009). Similarly, in most Spanish-speaking countries of Latin America, estimates of perinatal depression are around 36% (Lara et al., 2009). In Peru, between 24 and 40% of pregnant women suffer from perinatal depression, although only approximately 10% seek health services or access to care (Aramburu et al., 2008; Luna et al., 2009). Thus, perinatal depression produces a significant impact on public health due to its consequences on mental health in pregnant women and its potential negative impact on child developmental outcomes (Smith et al., 2020).

The emergence of COVID-19 as a global pandemic produced a significant mental health burden worldwide, and an increase in the manifestation of various psychological conditions, especially in groups considered to be at high risk, such as pregnant women (Iyengar et al., 2021; Khamees et al., 2021). In Peru, the social isolation measures ordered by the government aiming to reduce COVID-19 transmission and hospital burden for patients' care, as well as the problems derived from health emergencies and the economic uncertainty during the pandemic, may have caused situations of extreme stress, anxiety, and depression in pregnant women living in low-resource settings and who are more vulnerable psychologically (Bermejo-Sanchez et al., 2020). Additionally, fear related to the health of the newborn if the mother is infected with SARS-CoV-2, the uncertainty of the hospitalization process during the pandemic (Kotlar et al., 2021), and breastfeeding as a possible mechanism of COVID-19 infection (Zhu et al., 2020) may lead pregnant women to experience increased COVID-related distress. In this context, timely diagnosis of depression and mental health care should be a priority in pregnant women, since without adequate care the depressive condition may worsen or persist in the post-partum as observed in 50% of cases (Field, 2011), and even provoke negative outcomes for the mother and child such as low birth weight, intrauterine growth restriction, and other pregnancy complications (Rahman et al., 2007).

In 2015, the Peruvian Ministry of Health (MINSA) began to expand the delivery of mental health support interventions from tertiary-level systems to the primary care level in accordance with the Mental Health Gap Action Program (mhGAP) of the WHO (Miranda et al., 2017; Keynejad et al., 2018), and simultaneously with the passage of Law Regulation No. 29889 (modifying Article 11 of Law No. 26842) of MINSA, which guarantees the rights of people with mental health conditions (Toyama et al., 2017). One of the most vulnerable population groups toward which efforts should be directed to improve their mental health condition is the group of pregnant women in low-resource settings. Therefore, and in order to collaborate and support MINSA in the assistance and care of pregnant women, Socios En Salud (SES), a non-profit organization whose objective is to promote care for the most vulnerable populations to severe health conditions, has been developing through its Maternal and Child Health Program different interventions in pregnant women populations that include nutritional advice, health exams, facilitating referral to hospitals,

support for children's health (Miller et al., 2021), and a special emphasis on mental health care.

Because mental health is one of the key aspects in the well-being of pregnant women, SES, in collaboration with MINSA, implemented the WHO's "Thinking Healthy program (THP)" for the care and emotional support of pregnant women with psychological distress (Eappen et al., 2018). The THP is a low-intensity, community–based intervention that integrates cognitive behavioral techniques aiming to reduce the occurrence of depression during pregnancy. It includes different strategies that incorporate behavioral activation techniques, active listening, collaborating with the family, guided discovery, and homework. Its main advantage is that it that can be properly administered by trained non-specialists such as community health workers (CHW), or psychology students, among others, which facilitates its distribution in low- and middle-resource settings (Fisher et al., 2014; Turner et al., 2016; Eappen et al., 2018; Sikander et al., 2019). THP has been previously implemented in other countries with socioeconomic profiles like Peru where its efficacy has been demonstrated (Turner et al., 2016; Rahman et al., 2021). The aim of this article is to describe the results of a community-based mental health intervention by SES in a population of pregnant women in north Metropolitan Lima, which consists of a screening for depressive symptoms using the Patient Health Questionnaire (PHQ)-9 and implementation of THP support sessions, In two different contexts: before COVID–19 pandemic when activities were conducted in face-to-face modality, and during COVID-19 pandemic when activities were carried out remotely; and their re-evaluation at the end of the intervention.

## Materials and methods

### Study context

The SES Maternal & Child Health Program was implemented in 2018 in the district of Carabayllo in northern Metropolitan Lima, with approximately 330 thousand inhabitants, and consisted of strengthening the physical, nutritional, and mental health of pregnant women from the first trimester until 42 days post-partum and the health of their newborns. These activities were carried out through community-based educational and support interventions as a complement to the prenatal care that pregnant women received in MINSA health facilities. Also, in the third trimester of pregnancy, women were offered depression screening, and those who tested positive for depression were offered the WHO THP intervention for mental health support. These activities were conducted in face-to-face modality in the period before the COVID-19 pandemic (August 2018 to February 2020). In March 2020, the Peruvian government announced a national state of emergency due to the COVID–19 pandemic, establishing a total confinement to reduce transmission in the Peruvian population. Access to health facilities in Peru was also restricted during the COVID-19 pandemic to reduce staffing overload for medical care of COVID-affected patients. Mental health care, whose demand increased considerable during the COVID-19 pandemic, had to be re-designed to remote-delivery services, which also occurred in the SES' THP intervention.

### Study design

This study is a secondary data analysis that used datasets from two retrospective cohorts design comprised of third-trimester pregnant women (including adults and adolescents) from the Carabayllo

district in Lima who had a baseline screening for depression and subsequently received THP support sessions. The data corresponded to two intervention modalities: face-to-face (before the COVID-19 pandemic or pre-pandemic cohort) and remote (during the COVID-19 pandemic or pandemic cohort). Likewise, depression data was collected for all participants after THP sessions in both interventions.

### Study activities

The identified pregnant women were screened by SES psychologists, who were trained in the application of the PHQ-9 during sessions of approximately 30 min.

Although there are different depression screening instruments used in pregnant women, for this intervention the PHQ-9 was used in the validated version in Peruvian Spanish of the PHQ-9 tool (Calderon et al., 2012) because it allows the identification of different levels of depression, which facilitates differentiated care for the most severe cases. The severity of depression was classified according to PHQ-9 scores into mild (−9), moderate (10–14), moderately severe to severe (15–19) and severe (20 or more). A PHQ-9 cutoff score > 4 was used to classify pregnant women as positive for depression in order to capture as many participants as possible with depressive symptoms who were eligible for THP sessions. In addition, the PHQ-9 included a question related to suicidal ideation and/or self-harm. All participants who screened positive for depression were immediately contacted by telephone by SES psychologists who explained the screening results and offered THP support sessions. Participants who reported moderate–severe or severe depression or suicidal ideation were referred to mental health centers for specialized care and were also followed up by SES psychologists.

### Screening of depressive symptoms and assessment process
**Face-to-face screening.** The identified pregnant women were invited by the SES Maternal and Child Health program to the Mental Health program for the THP intervention. SES psychologists were contacted to explain about THP and to coordinate a home visit for screening. Depression-positive cases (scores >4) were invited to participate in the delivery of THP sessions by TCS.

**Remote screening.** In the pandemic cohort, pregnant women were identified by the EE.SS of Lima Norte and referred to the SES mental health program to participate in THP. Screening activities were conducted remotely by SES psychologists using the PHQ-9 through video calls on WhatsApp, using Google meet, or via phone calls. Participants who screened positive for depression were invited to receive remote THP sessions by TCS.

### Thinking Healthy program (THP)
**Face-to-face THP sessions.** In pre-pandemic cohort, THP sessions were administered in face-to-face modality and consisted of five modules, the first module included four sessions, and the remaining four modules included three sessions each. CHW received training sessions on THP delivery by SES psychologists. The THP sessions were held in the participants' homes weekly and lasted approximately 45 min. The participant was given a workbook and worksheets to help as a guide during sessions. SES psychologists also supervised CHW during sessions to verify that the intervention was carried out correctly and to support CHW's basic skills. After each session, the CHW completed a follow-up meeting sheet.

**Remote THP sessions.** In the pandemic cohort, THP sessions were reduced to only the first module, which consisted of four weekly sessions. Sessions were administered through WhatsApp video or Google Meet videocalls using cell phones for a better interaction between the participant and CHW. Occasionally there were connection problems that made video calls difficult, and the corresponding session was given through telephone calls. The participants physically received at home the workbook and worksheets that would later be used in THP sessions. At the end of the first virtual module, they were given an incentive in the form of vouchers for each session finished. During remote THP sessions, SES psychologists supervised the CHW's performance in the same call or videocalls but they were not visible. At the end of each session, SES psychologists called the CHW to provide feedback and complete the Virtual Encounter Form (VEF).

### Referral process
The referral process was carried out either in the pre-pandemic and pandemic cohorts if the enrolled participants reported moderate-to-severe or severe depressive symptoms, and/or suicide ideation/attempt. All women continued receiving THP sessions during the referral. The referral process included the psychologist identifying the facility closest to the participant's home, contacting that facility, and following up with the participant to ensure that the referral was successful for in-person or virtual care. In places where there was no access to a Community Mental Health Centre (CMHC), participants were referred to a nearby HC or hospital for acute needs. If participants with suicide ideation rejected their referral to mental health institutions, SES psychologists immediately contacted their relatives to inform them of the patient's condition and the importance of the referral to a specialized institution for mental health care. They were also provided with additional information on the closest facility.

### Supervisory activities
**Face-to-face supervision.** A total of 14 CHW provided face-to-face THP sessions and were trained and supervised by SES psychologists who also accompanied CHW during the first session of each module. Supervisory activities included weekly meetings to discuss the difficulties, limitations, feelings, alternatives, expectations, and challenges that arose in the CHW during THP sessions. Role-play, conflict resolution, and case discussion were used in these meetings.

**Remote supervision.** SES psychologists supervised each remote THP session, always informing the participant of his/her presence to avoid discomfort. During supervisory activities, the SES psychologists resolved possible doubts at the end of the session, and provided feedback to the CWH.

### Psychological re-assessment
After completing each module of the THP sessions, either in face-to-face or remote modality, participants were re-assessed in the same day using the PHQ-9 to determine if there was a reduction in baseline depression after THP sessions.

### Data collection and statistical analysis

Data from participants collected in both cohorts were uploaded for analysis into the SES computer system (SEIS) by psychologists. Sociodemographic variables (age categories, marital status, and degree of instruction) were obtained from participants enrolled in both intervention modalities (face-to-face and remote), whereas

other variables such a history of previous pregnancy, type of delivery (cesarean or natural), and place of birth (hospital or other place than hospital) were only obtained from participants in face-to-face intervention modality. PHQ–9 scores at baseline and at follow-up were also recorded. We described patients' characteristics and PHQ-9 scores using summary statistics as required. Percentages of pregnant women with depressive symptoms at baseline PHQ-9 screening (cutoff score ≥ 5) were reported for women participating in the periods before and during the COVID-19 pandemic, and bivariate comparisons between depressive symptoms and patients' characteristics were also performed. The severity of depressive symptoms, before and during the COVID-19 pandemic was also reported. We assessed the effects of THP sessions, delivered in either face-to-face or remote modality on reducing depressive symptoms, by comparison of screening and follow-up median PHQ-9 scores using the non-parametric Wilcoxon signed-rank test. Statistical analyses were carried out in Stata/SE 16.0. A significance level set to 5% was considered.

## Ethics

All the procedures described in this study comply with the ethical standards detailed in the Declaration of Helsinki and have been previously reviewed and approved by the Institutional Ethics Review Board (IRB) of the Universidad Peruana Cayetano Heredia (approval numbers 18,002 and 19,021).

## Results

### Participant characteristics

A total of 432 pregnant women were screened for depressive symptoms: 185 in pre-pandemic cohort (face-to-face intervention modality), and 247 in pandemic cohort (remote intervention modality). Median participant age was 27 years (interquartile range (IQR): 23–32), being the highest percentage between 21 to 40 years (83.6% (321/432)), 80.2% (324/404) were single, and 65.2% (264/405) had no higher educational level; there were no differences observed in these variables between the intervention modalities (all $p > 0.05$). Additionally, among the participants screened in face-to-face intervention, 71.3% (82/115) were multiparous, 62.1% (95/153) had birthed naturally, and 63.1% (99/157) gave birth in a hospital. Missing values were reported for marital status, degree of instruction, history of previous pregnancy, type of delivery, and place of birth (See Table 1).

**Table 1.** Depressive symptoms before and during the COVID-19 pandemic according to characteristics of study participants

| | Pre-pandemic cohort (face-to-face intervention) | | | | | | | Pandemic cohort (remote intervention) | | | | | | |
| | Total | | Depressive symptoms | | | | | Total | | Depressive symptoms | | | | |
| | (n = 185) | | No (n = 138) | | Yes (n = 47) | | | (n = 247) | | No (n = 130) | | Yes (n = 117) | | |
| Characteristics | Nº | (%) | Nº | (%) | Nº | (%) | $P^a$ | Nº | (%) | Nº | (%) | Nº | (%) | $P^a$ |
|---|---|---|---|---|---|---|---|---|---|---|---|---|---|---|
| Age (years) | | | | | | | | | | | | | | |
| 14–20 | 35 | 100.0 | 24 | 68.6 | 11 | 31.4 | 0.600 | 29 | 100.0 | 19 | 65.5 | 10 | 34.5 | 0.264 |
| 21–40 | 148 | 100.0 | 112 | 75.7 | 36 | 24.3 | | 213 | 100.0 | 109 | 51.2 | 104 | 48.8 | |
| 41–65 | 2 | 100.0 | 2 | 100.0 | 0 | 0.0 | | 5 | 100.0 | 2 | 40.0 | 3 | 60.0 | |
| Marital status[b] | | | | | | | | | | | | | | |
| Single | 141 | 100.0 | 109 | 77.3 | 32 | 22.7 | 0.139 | 183 | 100.0 | 104 | 56.8 | 79 | 43.2 | 0.027 |
| With couple | 37 | 100.0 | 24 | 64.9 | 13 | 35.1 | | 43 | 100.0 | 16 | 37.2 | 27 | 62.8 | |
| Degree of instruction[b] | | | | | | | | | | | | | | |
| Without higher education | 126 | 100.0 | 91 | 72.2 | 35 | 27.8 | 0.181 | 138 | 100.0 | 72 | 52.2 | 66 | 47.8 | 0.683 |
| With higher education | 52 | 100.0 | 43 | 82.7 | 9 | 17.3 | | 87 | 100.0 | 48 | 55.2 | 39 | 44.8 | |
| Previous pregnancies[b] | | | | | | | | | | | | | | |
| No | 33 | 100.0 | 30 | 90.9 | 3 | 9.1 | 0.116 | – | – | – | – | – | – | |
| Yes | 82 | 100.0 | 63 | 76.8 | 19 | 23.2 | | – | – | – | – | – | – | |
| Type of birth[b] | | | | | | | | | | | | | | |
| Cesarean | 58 | 100.0 | 47 | 81.0 | 11 | 18.9 | 0.332 | – | – | – | – | – | – | |
| Natural | 95 | 100.0 | 70 | 73.7 | 25 | 26.3 | | – | – | – | – | – | – | |
| Place of Birth[b] | | | | | | | | | | | | | | |
| Place other than hospital | 58 | 100.0 | 43 | 74.1 | 15 | 25.9 | 1.000 | – | – | – | – | – | – | |
| Hospital | 99 | 100.0 | 74 | 74.8 | 24 | 25.3 | | – | – | – | – | – | – | |

[a]Fisher exact test.
[b]Missing values were reported for marital status (n = 7 in face-to-face modality and n = 21 in remote modality), degree of instruction (n = 7 in face-to-face modality and n = 22 in remote modality), previous pregnancy (n = 70 in face-to-face modality), type of birth (n = 32 in face-to-face modality) and place of birth (n = 28 in face-to-face modality).

## Screening for depressive symptoms

In pre-pandemic cohort the frequency of depressive symptoms (PHQ-9 score ≥ 5) among pregnant women was 25.4% (47/185), whereas in pandemic cohort the frequency rose nearly twofold to 47.4% (117/247). Before COVID-pandemic, depressive symptoms were not different between age categories ($p = 0.600$, exact), but were higher (although not statistically significant) in pregnant women living with a couple ($p = 0.139$) and in pregnant women with higher educational level ($p = 0.181$, Table 1). During the COVID-19 pandemic, depressive symptoms were not different between age categories and degree of instruction, although depressive symptoms were significantly higher in women living with a partner couple compared to single women (62.8% (27/43) and 43.2% (79/183), $p = 0.027$).

According to severity of depressive symptoms, in the pre-pandemic cohort a higher percentage of screened participants reported mild and moderate depressive symptoms (16.2% (30/185) and 7% (13/185)), two women reported moderately severe depressive symptoms (1.1%), and two other cases had severe depressive symptoms (1.1%). In the pandemic cohort, the percentages of mild depressive symptoms increased considerably among screened participants (42.2% (105/247)), whereas moderate and moderately severe depressive symptoms were observed in 10 (4.0%) and 2 cases (0.8%), respectively, and no cases with severe depressive symptoms were reported (see Table 2).

## THP support sessions

Of the 432 participants initially screened, 38% met the criteria to receive the THP sessions, because they did show evidence of depressive symptoms, both in the pre-pandemic cohort (n = 47) and pandemic cohort (n = 117). Of this group, 48 people did not complete the sessions, the main reason being that they dropped out of the intervention (11 in the face-to-face modality and 24 in the remote modality). PHQ-9 re-assessment was conducted only in participants who completed the intervention in their last session. There were 27 participants who completed the five face-to-face THP module sessions before COVID-19 pandemic, and 89 participants who completed the first remote THP module during COVID-19 pandemic. Pre- and post analysis showed a significant reduction in median PHQ-9 scores after THP sessions either in face-to-face modality (median PHQ-9 score changed from 8 to 2, $p < 0.001$) and remote modality (median PHQ-9 score changed from 7 to 2, $p < 0.001$, Table 3).

## Discussion

In this retrospective cohort analysis of depression among pregnant Peruvian women before and during the COVID-19 pandemic, we found a near two-fold increase in depressive symptoms (from 25.4% to 47.4%), which is in line with previous in Peruvian and other Latin American studies (Barzola, 2021), and corroborates the vulnerability of pregnant women to develop depression, especially in high-stress situations as occurred during the COVID-19 pandemic (Mei et al. 2021). Our results also show that THP sessions, delivered either in face-to-face or remote modality, reduce depressive symptoms among pregnant women, and should be evaluated in future interventions in other contexts.

Depressive symptoms have been reported to be high among pregnant women in prior studies in low-income endemic settings (Humayun et al., 2013; Biratu and Haile, 2015; Thompson and Ajayi, 2016). Likewise, previous studies have reported an increase of depressive symptoms among pregnant women (Ayaz et al., 2020; Lebel et al., 2020) as occur in our study. However, it is possible that the high percentage of perinatal depression observed in our study could be due to the use of a more sensitive PHQ-9 cutoff score (≥5) in comparison with previous studies of depression in pregnant women that use higher cutoff scores (≥10), which are more specific, but less sensitive (Sidebottom et al., 2012; Woldetensay et al., 2018).

In our study, age was not associated with depressive symptoms in pregnant women, both before and during the COVID-19 pandemic. These results are contradictory with previous studies that showed high depressive symptoms in younger women (≤25 years), mainly due to less favorable economic conditions, lower educational level, and/or lower salaries (Kheirabadi and Maracy, 2010; Bodecs et al., 2013). It is likely that in our study, many young pregnant women during the COVID-19 pandemic had returned to their parents' home and had received better social support, which may protect those typically exposed to the risk of low-quality living conditions. Similarly, it is possible that the older pregnant women in our study were affected by the stress conditions generated by the COVID-19 pandemic, such as an increasing fear of becoming infected during assistance or cesarean section in childbirth, which may have decreased the rate at which older pregnant women sought support in these settings pre-pandemic. Surprisingly, depressive symptoms in pregnant women were higher in participants who reported living with a partner than in single women – with a statistically significant difference identified during the COVID-19 pandemic.

Many studies showed that pregnant women who experience depression are more likely to be single (Adewuya et al., 2007; Jeong

**Table 2.** Severity of depressive symptoms according to the PHQ-9 before and during the COVID-19 pandemic

| Severity of depressive symptoms | Scores | Total (N = 432) | | Pre-pandemic cohort (N = 185) | | Pandemic cohort (N = 247) | |
|---|---|---|---|---|---|---|---|
| | | n | % | n | % | n | % |
| None | 0–4 | 268 | 48.5 | 138 | 74.6 | 130 | 52.2 |
| Mild | 5–9 | 135 | 24.4 | 30 | 16.2 | 105 | 42.2 |
| Moderate | 10–14 | 23 | 4.2 | 13 | 7.0 | 10 | 4.0 |
| Moderately severe to severe | 15–19 | 4 | 0.7 | 2 | 1.1 | 2 | 0.8 |
| Severe- | ≥ 20 | 2 | 0.4 | 2 | 1.1 | 0 | 0.0 |

**Table 3.** Comparisons of PHQ-9 scores for depressive symptoms at baseline and after implementation of THP support sessions in pregnant women before and during the COVID-19 pandemic

| PHQ-9 scores | Baseline | | | | | Re-assessment | | | | |
|---|---|---|---|---|---|---|---|---|---|---|
| | N | Median | Min | Max | IQR | Median | Min | Max | IQR | $P^a$ |
| Type of intervention | | | | | | | | | | |
| Face-to-face[b] | 27 | 8 | 5 | 14 | 4 | 2 | 0 | 7 | 2 | < 0.001 |
| Remote[c] | 89 | 7 | 5 | 17 | 2 | 2 | 0 | 12 | 2 | < 0.001 |

[a]Comparisons were performed using Wilcoxon sign-rank test.
[b]Included the five THP module sessions.
[c]Only included the first THP module session.

et al., 2013). While the partners could provide emotional support to depressive women, troubled relationships characterized by stress such as those occurring during the COVID-19 pandemic in many families with low resources, and gender-based violence may also place an additional burden of stress for pregnant women, making it more difficult for them to bear pregnancy (Inter-Agency Standing Committee, 2015; Biaggi et al., 2016; Mittal and Singh, 2020; Opanasenko et al., 2021). We also observed higher percentages of depression in women with a previous history of pregnancy and natural childbirth before the COVID-19 pandemic, similar to previous studies (Golbasi et al., 2010; Fisher et al., 2013; Redshaw and Henderson, 2013), although we do not know if this trend is also observed in our population during the COVID-19 pandemic.

The results of our intervention before COVID-19 pandemic showed a significant reduction in PHQ-9 scores in pregnant women who completed the five face-to-face THP module sessions, which was similar to findings from other studies conducted in few human resource settings (Vanobberghen et al., 2020). These findings also demonstrate the role of CHWs in administering the THP sessions in the community. Unlike other interventions for depression in pregnant women that include educational psychotherapies or interpersonal therapies performed only by psychologists, THP sessions use cognitive behavioral therapy methods that can be administered by trained non-specialists such as CHWs, which improves its cost-effectiveness, and allows a greater distribution among the community (Rahman et al., 2021). Our findings of THP sessions in reducing depressive symptoms remained significant during the COVID-19 pandemic, which suggest that remote delivery of THP sessions is possible during pandemics that introduce strict lockdown policies. It is also important to note that the reduction of follow-up PHQ-9 scores was significant after a single remote THP module during the COVID-19 pandemic. Future studies, including in other settings with vulnerable populations, are required to corroborate these findings. Additionally, improving the dissemination of THP for emotional support in pregnant women using technologies such as chatbots and training of CHWs for its administration could improve enrollment, adherence, and intervention outcomes.

Our study has limitations. Due to the non-probabilistic nature of the sampling in our study, the results may be biased and not representative of the entire population. The presence of missing values in some covariates and the absence of some clinical patients' characteristics related to pregnancy, both in face-to-face and remote interventions, limit our ability to explain in more detail the characteristics of depressive symptoms in the study population. Also, we cannot assume a direct effect of the delivery of THP sessions in reducing depressive symptoms observed at re-assessment, since it is also possible that follow-up PHQ-9 scores naturally decreased over time, considering the stages of the last trimester of gestation and postpartum, where physical and emotional changes are evident, as well as the change of roles of the mother and the variation in the stressful events she goes through (Rallis et al., 2014; Răchită et al., 2022; Wegbom et al., 2023). Therefore, the inclusion of a control group without the intervention in a quasi-experimental design could improve the design of the intervention and future controlled studies may help to demonstrate the effectiveness of THP sessions for perinatal depression. The lack of information about depression assessment and adherence of participants after each THP module during face-to-face intervention does not allow assessing the behavior of participants over time during the intervention. Finally, further studies implementing THP sessions in pregnant women, either in face-to-face or remote modalities, should also include its effect on birth and child outcomes.

Depressive symptoms have increased considerably in pregnant women from vulnerable settings in Lima during the COVID-19 pandemic, so timely diagnosis and delivery of THP sessions in community-based interventions could help reverse this condition. Our findings demonstrate a high percentage of depression in pregnant women, with a higher percentage of cases reporting mild or moderate depressive symptoms; and, appropriate mental support, such as THP sessions (either in face-to-face or remote modality), can help to reduce depressive symptoms and thus improve the mental health status of affected women. Furthermore, our findings also suggest that a remote intervention with an abbreviated version of THP (first module only) may improve the health status of pregnant women with depressive symptoms immediately after the intervention delivery. However, further studies are needed to determine whether this effect is seen over a longer period of time, such as one or three months post-intervention.

**Open peer review.** To view the open peer review materials for this article, please visit http://doi.org/10.1017/gmh.2023.45.

**Acknowledgements.** Gratitude is expressed to the population of the Carabayllo district, Lima, Peru, and MINSA for their support in carrying out this study. We are grateful to the individuals who participated in the studies in each of the EQUIP sites and to all members of the EQUIP team for their dedication, hard work, and insights. We also thank the professionals Karen Ramos and Milagros Dueñas for their great work in the related projects. Thanks to the IMPACT PIH team for their support in publishing this article. The authors alone are responsible for the views expressed in this article and they do not necessarily represent the views, decisions, or policies of the institutions with which they are affiliated.

**Financial support.** This study was supported by the *WHO EQUIP* initiative provided by USAID 2020/1043477–0, Partners in Health (PIH), and Grand Challenge of Canada (GCC) (R-TTS-2106-40,386) Global Mental Health: The

PIH Cross-Site Mental Health Learning Collaborative: Capacity building for mental health care delivery and implementation. This research was funded by the NIHR using aid from the UK government to support global health research. The views expressed in this publication are those of the author(s) and not necessarily those of the NIHR or the UK government.

**Competing interest.** The authors declare no competing interests exist.

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
