## [Reviewer Report]

Perinatal depression is a mental health problem that fluctuates between 24% and 40% and that not only significantly affects maternal mental and physical health, but also the newborn in physical, mental, and cognitive aspects, with serious repercussions. in adult life. Therefore, the need to cover mental health care for this population is essential.

To strengthen the Peruvian health system, Socios En Salud, implemented the WHO Healthy Thinking program with the aim of reducing symptoms of depression in pregnant women in the community and with the accompaniment of a Community Health Agent (ACS) trained in the North of Lima Peru. In the implementation, the intervention was completed in 116/432 pregnant women in the third trimester, before and after isolation by COVID 19.

Before the isolation due to COVID 19, the ACS carried out the 16 sessions in the homes of pregnant women. And, during the isolation by COVID 19, the first module (4 sessions) was implemented remotely. In both interventions, the ACS received supervision and follow-up from mental health professionals.

The program managed to benefit pregnant women and ACS. Pregnant women receiving follow-up sessions by ACS and Mental Health personnel, reducing levels of depression. While the ACS benefited from training in the Healthy Thinking program and strengthened their skills in managing depression during maternity.

The results of the investigation show that the remote and face-to-face intervention managed to reduce the symptoms of depression in pregnant women. Likewise, it generates the opportunity to discuss possible modifications to the session program and to continue investigating the effectiveness and quality of low-intensity psychosocial interventions delivered in low-resource countries. Another long-awaited result was the scaling of the program to the public sector, including THP to the Maternal Mental Health Program to health professionals and is being led by the Ministry of Health.

---

## [Reviewer Report]

Perinatal depression is an important public health condition and efforts to address it in Peru through primary care provide opportunities to learn important implementation lessons. This study reports two interesting findings. First, that the prevalence of perinatal depression almost doubled during the COVID pandemic. Second, that the provision of the Thinking Healthy intervention remotely was effective in reducing the symptoms in affected mothers. The study uses routine data in a pre-post design, and the limitations of the study are covered in detail in the discussion section. However, the paper would benefit from a more detailed description of the ‘remote’ modality of treatment offered during the pandemic. For example, were the sessions delivered through mobile phones or internet? A comment on the availability of such modalities to the patients, and suggestions to overcome any ‘digital divide’ would be useful.

---

## [Reviewer Report]

The submitted manuscript ‘Perinatal depression and implementation of the “Thinking Healthy Program” support intervention in an impoverished setting of Lima, Peru: assessment before and during the COVID-19 pandemic’ provides a compelling account of the implementation of this evidence based perinatal depression intervention, successful adaptation of the intervention into a remotely delivered and abbreviated format, and meaningful improvement in depression symptom severity over the course of the program for participants who could be reassessed.

This manuscript details the successful delivery of THP in an impoverished metropolitan setting in Peru, both through the traditional face-to-face approach well-validated in other settings as well as through a novel remote approach utilizing only the first of five THP modules. The study was limited by a large proportion of participants lost to followup, lack of controls in study design, and lacking assessment of duration of benefits, opacifying the true effectiveness of the intervention. Despite these limitations, this study’s results signal the possibility of substantial benefit of THP to pregnant individuals with prenatal depression in the Peruvian context.

The manuscript would primarily benefit from further efforts to account for the participants who were not able to be reassessed. This could include discussion of why reassessment is missing, how many sessions of the intervention were completed by those who were and were not reassessed, and whether there were notable trends in demographics or baseline depression between these groups.

As discussed by the authors, future research would benefit greatly from a controlled study design. The ethical and practical challenges of this are not to be understated, especially given that 1) screening itself and health system referrals received by a control group may or may not be already beyond usual care, 2) usual care may still be inadequate to support control group, and 3) THP is designed to be started pre-partum whereas any waitlist control group crossover to intervention would necessarily be post-partum. That said, it is important to reveal to what extent improvement in perinatal depression is due to the intervention versus the natural course of depressive symptom severity in the perinatal period (regression to the mean). Perhaps individuals screened into the study who did not consent to participate in the THP intervention would however wish to consent to a single followup assessment, at least serving as non-random controls. Additionally, followup 1, 3, or 6 months after completion of the intervention would provide additionally valuable data about the duration of benefits. Funding to support these elements of study design may be essential to demonstrating whether this promising intervention is truly effective enough to merit scale-up.

Major comments requiring attention:

1) Risk of attrition bias: There is limited discussion of the large proportion of women not included in reassessment, which could introduce substantial bias (i.e. if women with elevated baseline PHQ-9 or THP non-responders were more likely to drop out of study). Authors should state what factors led to not being included in reassessment and what percent of planned sessions were attended by those who were and were not reassessed. Demographics and baseline perinatal depression severity should be directly compared between participants reassessed versus not reassessed.

2) Duration: Authors should comment in discussion about the limited ability to assess duration of benefits in this study, which may be particularly relevant to interpreting the success of the novel remote and abbreviated approach to THP. The manuscript should state the median time after completion of the final session that re-evaluation occurred.

3) Regression to mean: As the authors discuss in the conclusion (p10), PHQ-9 improvements could reflect the natural course of perinatal depression or changes to external circumstances (e.g. related to the pandemic and its socioeconomic consequences) rather than the THP intervention. In their discussion of this limitation, authors should discuss how to interpet the literature’s estimates of the improvements expected by the passage of time or usual care to contextualize their results.

For example, a recent pooled analysis of THP RCTs in India and Pakistan demonstrated improvement from PHQ-9 prepartum baseline median of 14 to followup median of 6.0 in controls receiving enhanced usual care and 5.1 in intervention group, suggesting some but not predominant effect of THP in improved symptoms (Vanobberghen et al., 2020). However, enhanced usual care in that study may be more effective than the usual care available in the context of the submitted manuscript. In contrast, an older RCT of interpersonal psychotherapy for postpartum depression showed much more modest improvement in waitlist control group compared to intervention group (Beck Depression Inventory 19.8 to 16.8 vs 19.4 to 8.3, respectively; O’Hara et al., 2000), suggesting postpartum depression is not self limited, though the baseline was notably assessed postpartum. The original RCT for THP demonstrated 47% of participants no longer met depression criteria among controls (receiving equal number of community health worker visits without specific THP intervention) versus 77% in the intervention group at 6 months (Rahman et al., 2008).

The authors may be able to find the best examples of progression of depression symptom scores in a control group of a perinatal depression intervention that is representative of usual care available to their participants and the timing of this study’s baseline and reassessment. This would contextualize how to interpret the role of the THP intervention in the substantial symptom improvement apparent from their results.

Additional minor comments:

1) Citation needed for “prevalence estimates of perinatal depression range from 7 to 20%, although it can be as high as 35 to 50% in low-and-middle income settings” or this range should clearly relate to the figures subsequently cited (p1).

2) If the Edinburgh Postnatal Depression Scale was not assessed, authors should describe why PHQ-9 was preferred. If the former was assessed, these data should be presented alongside PHQ-9.

3) “... which are more specific, but less specific” should likely state “... which are more specific but less sensitive” (p9).

Remarks on copy editing errors are not included in this review.

Given that the authorship team is primarily based in Peru, please have these comments translated into Spanish if possible. If this is not possible, this reviewer can provide an unprofessionally translated version of these comments.

References:

O’hara, M.W., Stuart, S., Gorman, L.L. and Wenzel, A., 2000. Efficacy of interpersonal psychotherapy for postpartum depression. Archives of general psychiatry, 57(11), pp.1039-1045.

Rahman A, Malik A, Sikander S, Roberts C, Creed F. Cognitive behaviour therapy-based intervention by community health workers for mothers with depression and their infants in rural Pakistan: a cluster-randomised controlled trial. Lancet. 2008 Sep 13;372(9642):902-9. doi: 10.1016/S0140-6736(08)61400-2. PMID: 18790313; PMCID: PMC2603063.

Vanobberghen F, Weiss HA, Fuhr DC, Sikander S, Afonso E, Ahmad I, Atif N, Bibi A, Bibi T, Bilal S, De Sa A, D’Souza E, Joshi A, Korgaonkar P, Krishna R, Lazarus A, Liaqat R, Sharif M, Weobong B, Zaidi A, Zuliqar S, Patel V, Rahman A. Effectiveness of the Thinking Healthy Programme for perinatal depression delivered through peers: Pooled analysis of two randomized controlled trials in India and Pakistan. J Affect Disord. 2020 Mar 15;265:660-668. doi: 10.1016/j.jad.2019.11.110. Epub 2019 Nov 23. PMID: 32090783; PMCID: PMC7042347.

---

## [Reviewer Report]

It is a report of an interesting program and one that could be useful for its adaptation in other sites with scarce resources. However, it has strong limitations to reach valid conclusions.

- The purpose of the study is not clear. Are two delivery modalities of the THP program being compared? Is it an effectiveness evaluation?

- The use of the PHQ-9 for perinatal depression screening must be justified, considering that there are own instruments (such as the Edinburgh Postnatal Depression Scale, EPDS, which has also been validated for depression screening in pregnancy).

- The use of a low cut-off for the PHQ-9 (>4) must be justified for the identification of possible cases, since this creates a risk of misclassifying cases. What are the perinatal depression screening values for this cut-off?

- The first time an acronym is used, it must be indicated what it refers to. See CMHC in 2.3.3.).

- There is no control group and only the change in the intensity of the symptoms (PHQ-9 score) is reported, and the placebo effect of participating in a group cannot be ruled out. Therefore, it cannot be considered an evaluation of the effectiveness of the THP program.

- In addition, to evaluate results, the scores are compared only in women who have undergone pre- and post-intervention evaluation, without considering the biases that may exist by not considering those who do not have measurement in the post-intervention time.

---

## [Reviewer Report]

The authors have addressed the concerns raised in the review process. I am satisfied with the quality of the work and recommend publication.

---

## [Reviewer Report]

The authors responded one by one to the comments, clearly and precisely. They also made changes to the manuscript, based on the comments, which greatly improved its quality. It is recommended to accept to publish.